# An Understanding of Christians' Roles in Human Migration through the Biblical Theme of Shamar: From Genesis to the Good Samaritan

Paul C. Fong

Sacred Heart Preparatory, Atherton, CA 94301, USA; pfong23@shschools.org

**Abstract:** The complex and urgent migration crisis demands a proper framework to formulate and drive the appropriate solutions. In this study, the author attempts to present a theological framework on the roles of Christians to migrants grounded in the theme of "shamar" (שָׁמַר; keep), tracing it through the Bible, from the Old Testament to the New. We argue that welcoming (keeping) strangers originated from man's first mission by exegeting Genesis 2:15 and 4:9 (the primordial assignment in the creation narrative and the denial in the first murder). After the Fall, the task is passed to all descendants and is then explicitly included in the Law. In the New Testament, "shamar" and its meanings are further revealed through three characters: the Samaritan leper, the Samaritan woman, and the Good Samaritan. The Samaritans show us that one should see a migrant as not only a brother or sister but also as oneself and as a part of one's mission. One should see with an open heart and be ready to be converted. A Samaritan may not be just one who comes from Samaria; instead, one who practices "shamar"—welcomes strangers unconditionally.

**Keywords:** immigration; Samaritans; the Good Samaritan; shamar; the Bible; the Old Testament; the New Testament; exegesis

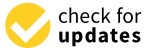



## 1. Introduction

"The Age of Migration", a phrase coined by Peter Phan, fittingly describes the scale of human mobility at the current time (Phan 2016). The significant movement of migrants and refugees across national borders has become one of the defining characteristics of the world today. There were around 281 million international migrants in the world in 2020, which equates to 3.6 percent of the global population (IOM 2022), compared to 2.8 percent in 2000 and 2.3 percent in 1980. The dramatic migration increase in recent years presents a mounting crisis in many ways. While many individuals migrate out of choice, others migrate out of necessity. According to the UNHCR, the number of globally forcibly displaced people was 89.3 million at the end of 2021. Of these, 27.1 million were refugees. In addition, 53.2 million were internally displaced, and 4.6 million were asylum seekers (UNHCR 2023).

Regarding the effect migrants have on the countries they pass through and the destination country, any positives and negatives will surely multiply as more migrants make the dangerous journey. These migrations can cause various issues, including caring for migrants if they are injured or cannot continue their journey. Once migrants reach their destination country, they either face a long process of achieving legal asylum or make the decision to cross illegally. For the migrants that enter the country, legally or illegally, there are additional challenges. States and cities face additional costs in maintaining and expanding services to meet the growing population resulting from this migration.

Such a scale of migration has significant impacts on both the people and the places of the migrants' origin and destination. When supported by appropriate policies, migration can contribute to inclusive and sustainable development in both the origin and destination countries and benefit the migrants and their families (United Nations Department of

Economic and Social Affairs, Population Division 2020). However, the opposite could happen when there is a lack of well-intentioned government policies and proper perspectives from people on both sides. Despite the complexities, institutions, and governments are formulating a concrete response to the current situation.

*Theology and Migration Survey*

The issue of immigration is complex and requires a correspondingly complex response informed by many disciplines. Before considering legislative or technological solutions, it is vital to consider the disposition and mindset needed to confront this challenge. In the last two decades, many scholarly works on the theological investigation of migration have been developed. A critical analysis of the recent study on theological engagement with migration can be found in Ilsup Ahn's paper (Ahn 2019). The two notable research are Phan's Deus Migrator, presenting God-On-The-Move and the Church as an Institutional Migration, and Daniel Groody's immigration theology, focusing on the spirituality of migrants and the ecclesial justice for refugees. In addition, Groody and Campese edited a volume of 17 essays from an international body of theologians to present different theological perspectives on the topic (Groody and Campese 2008).

So far, most of the works can be broadly categorized into three groups: first, focus on the spiritual and pastoral role of the Church in the phenomenon of human migration (Phan 2016). Second, use the life of Christ, the Old Testament Jewish movements, or the early Christian movements to reflect on the spirituality of the migrants (Groody 2022; Baggio 2005). Third, exegete the final section of Matthew 25 (the eschatological discourse) to investigate the proper Christian response (Groody and Campese 2008).

The author argues that additional research on the roles of an individual Christian based on the mission of man is necessary to create a proper Christian mindset. In addition, the author argues that the unique characteristics of migrants and refugees are underpresented with a collective analysis using the final section of Matthew 25.

This study seeks to present a theological discussion for Christians to understand their roles, focusing on why, how, and so what. Why should Christians play a role? How to respond? So what? The three goals of this research are: first, to demonstrate using two Biblical texts—Genesis 2:15 and 4:9—that welcoming foreigners is a part of the mission of man; second, to illustrate the concept of unconditional welcome using the three Samaritans in the New Testament and its transforming power based on Joseph Ratzinger's analysis (Ratzinger 2007); third, to discuss how to formulate such a mindset in the current situation.

## 2. Method: Narrative-Critical Biblical Reflection Based on the Hebrew and Greek Texts

This study employs the narrative-critical biblical reflection approach described in Ahn's work (Ahn 2019). With migration being such a deep-rooted concern throughout the Salvation History, one can look to scripture to find one's response. We begin with a critical contextual analysis of selected Bible verses in their original languages and then formulate a synthesis based on the theme of this research. After that, we compare our study with well-known commentaries.

This paper focuses on the theme of שָׁמַר ("shamar"). "Shamar" is a Hebrew word that can be translated into various meanings. The most common ones are "to keep", "to watch", and "to preserve." Several notable instances of this word tie the mission of man and the theme of migration in the Bible. Capitalizing on the relationship, we develop a proposal for the Christian response to migration. Section 3 argues that welcoming (keeping) strangers is a part of man's mission by dissecting two Biblical texts—Genesis 2:15 (the primordial assignment in the creative narrative) and Genesis 4:9 (the denial in the first murder narrative).

In the New Testament, Samaritans play the opposite of their role in the Old Testament. They are no longer portrayed superficially as enemies of the Jewish people but rather friends of the newly formed Christians. Early Church writers, such as Jerome of

Stridon, argue that Samaritans call themselves *shamerim*, the guardians or keepers of the Law, rather than *Shomronim*, Samaritans, as they are otherwise known to us (Crown 1989, p. 196). Section 4 begins with a review of the history of Samaritans. Then it is followed by the exegesis of three Samaritan encounters. In their encounters with a stranger, each Samaritan is inwardly converted and then offers an unconditional welcome to the stranger.

Section 4 offers suggestions on how to apply the "shamar" mindset on the pastoral level. Section 5 presents the conclusion of this research.

*2.1. The Theology of Migration Framework*

As shown in Figure 1, Groody proposes a framework for developing a migration theology (Groody, A theology of migration: the bodies of refugees and the body of Christ 2022). The theological exploration of migration interweaves three interrelated levels of action and contemplation: (1) the pastoral level, (2) the spiritual level, and (3) the theological level.

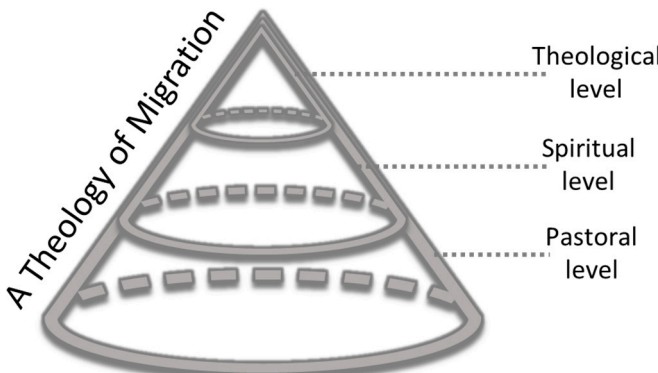

**Figure 1.** A Theology of Migration Framework.

This research utilizes the above framework to investigate the role of Christians in the phenomenon of migration. Each section begins with an exegesis of a Biblical text in its original language and then interprets its meanings in the context of migration. It follows with a spiritual study and a suggestion for pastoral action.

*2.2. The Bible Translations*

Since this study is based on Hebrew and Greek words and their meanings, choosing the appropriate manuscripts and translations is critical to the accuracy and success of the research. The Leningrad Codex is selected for studying the Old Testament in Hebrew because of its historical and religious significance (Kahle 1960), while Textus Receptus (TR) is used for studying the New Testament in Greek because many popular English translations are based on TR, including the King James Version. For interpreting the English text, the King James Version (KJV) is used because of its cultural familiarity and the availability of a concordance—Strong's Concordance. English Standard Version (ESV), an "essentially literal" translation, is used to provide a second opinion in analyzing the English text because of its precision in the English translation of the original text (About the ESV 2023). In addition, Septuagint (LXX) and Vulgate assist the interpretation of the Hebrew words in the Old Testament. The Greek and Latin translations of the Old Testament provide insights into the understanding of the early Christians and the Catholic Church.

**3. The Old Testament: The Mission of Man—שׁמר ("Shamar")**

The primary focus of this section is to identify man's mission through one of the two initial tasks he is given during creation and how men have performed this task in the Old Testament.

### 3.1. Genesis: The Book of Creation, the Book of Migrations

*"In the beginning, . . . ."*

The Book of Genesis is the first book of the Bible and is often referred to as the book of "origins." Its title in English, Genesis, comes from the Greek word "Γένεσις", which means birth, lineage, and generation. It is the most important book of the Bible and introduces key foundational themes for the rest of the Bible (Fox 1983, pp. 3–7). Migration is one of these. It provides theological prototypes and spiritual archetypes related to migration. That is why Andrew Walls, a historian of missions, called Genesis the Book of Migrations (Walls 2002). In addition, the first three chapters provide clues on the design and purpose of humankind.

### 3.2. Genesis 2:15—Shamar (Keep) the Gan (Garden)

*"And the LORD God took the man, and put him into the Garden of Eden to dress it and to keep it."*

Man is unique, and his creation differs from the rest (Ennis 2014, pp. 44–45). Man is created on the last day, the climax of creation; at the conclusion of man's creation, God noted, "*it was very good.*" God's purpose in creating man is stated in Genesis 1:26, "*let them have dominion . . . .*" Immediately after the creation narrative, the first man is placed in the Garden of Eden and is given two tasks: to dress (עבד) (abad 2023) and to keep (שמר) (shamar 2023). The Hebrew word שמר ("shamar") appears 468 times in the Old Testament, while the Hebrew word for dress ("abad") appears only 289 times. Shamar's primary meanings are to keep, watch, and preserve (Strong 1890, p. H8104). In the KJV, it is translated into 12 different English words depending on the passage's context. Before investigating the meaning of "shamar" in the Bible verse, it is vital to understand the context of the first two chapters. The English word "garden" is used for the Hebrew word גן (gan). גן has two primary meanings: garden and enclosure. All English translations chose the word "garden." In addition, Eden (עדן) means paradise. It is translated into the Greek word "Εδέμ" in the Septuagint. Therefore, it makes sense to translate "shamar" into keep since the term could be related to plant and garden maintenance.

Despite the high occurrence, this combination of the two verbs is not used elsewhere in the Pentateuch until the same two words are found in a description of the duties of the Levitical priests in the Tabernacle (Numbers 3:7–8; 8:26; 18:7). Thus, another meaning is that the chapter also describes Adam as a priestly figure, commissioned to serve in the Garden of Eden (Bergsma and Pitre 2018, p. 103). A priest functions as a mediator between God and creation. However, Adam is more. Bergsma calls him the vice-regent of all creation. He names the animals, which implies ownership. If the Levitical priesthood is to mimic the primordial priesthood, one of his tasks would be to make the sacrifice to God: קרבן (an offering, 284 times) and זבח (a sacrifice, 162 times). The general idea of sacrifice is to present a "gift" to God (Harris et al. 1980). Since sin has not entered the world, the purpose of the sacrifice would be to express gratitude and love for God.

Rydelnik and Vanlangham draw a similar conclusion using a different approach: the Hebrew linguistic analysis (Rydelnik and Vanlaningham 2014, pp. 41–42). First, they recognize that "it" is mistakenly understood as an ending attached to the verbs that function like a pronoun. Even if a pronoun ending must be used, it would be feminine and better translated as "her" than "it." In the previous verses, "garden" is masculine as its antecedent. Second, the ending that only looks like a "her" or "it" pronoun ending is nothing more than a typical ending for Hebrew infinitives without being a pronoun ending. Third, they suggest that the infinitives would be better translated as to worship (or "to serve") and to obey (or "to keep charge"). They also point out that Genesis 2:15 presents the only purpose statement in the first two chapters—the Ideal Responsibility of Man. They conclude that the purpose for which man is created is to worship and obey God. This obedience

in worshipping and obeying God is expressed by keeping his command in verse 16. The following sections explore how keeping God's commands becomes keeping others.

### 3.3. Genesis 3—The Fall and the Exile (the Loss of the Mission and the First Forced Migration)

> "Therefore the LORD God sent him forth from the Garden of Eden, to till the ground from whence he was taken. So he drove out the man; and he placed at the east of the Garden of Eden Cherubims, and a flaming sword which turned every way, to keep (שמר) the way of the tree of life."
>
> Genesis 3:23–24 KJV

Unable to resist the serpent's temptation, the man and the woman break the Law and eat the fruit from the tree of the knowledge of good and evil. Immediately, they become ashamed of their nakedness and try to hide from God. Unable to confess and repent of their wrongdoing, they are banished from the Garden of Eden, and the man is condemned to work to find what he needs to live. Nevertheless, God is still concerned for Adam and Eve's welfare, replacing their inadequate leaf garments with "coats of skins" (Genesis 3:21). It implies that God kills an animal (maybe for the atonement for the first sin) and takes away Adam's priestly role.

In the following eight chapters of the Book of Genesis, disobedience grows and metastasizes. The next section investigates the brokenness of the relationship between the first brothers. It results in murder—the first sin after the First Sin—a further deviation from "shamar".

### 3.4. Genesis 4:9—No Longer a Keeper

> "And the LORD said unto Cain, Where is Abel thy brother? And he said, I know not: Am I my brother's keeper?"
>
> Genesis 4:9 KJV

Then the first couple has two sons: Cain and Abel. The first activity described is making an offering to God, which they may have learned from their father. However, this first sacrifice results not in a renewed relationship with God, but rather a further shattering through sin. Out of jealousy, Cain kills Abel. When confronted by God, Cain gives a flippant response: "Am I my brother's keeper (השמר)?" His grudging reaction implies that he knows about the task of "shamar" and his mission. However, he does not live up to the keeper role and does the extreme opposite of "shamar." When interrogated by God, Cain lies. He is worse than the first sinners, who sought merely to shift the blame. The punishment of Cain is more severe than that of the first parents. He is banished from the soil, cursed to be a wanderer—further forced migration.

### 3.5. The Law and Commandments

> "And I prayed unto the LORD my God, and made my confession, and said, O Lord, the great and dreadful God, keeping (שמר) the covenant and mercy to them that love him, and to them that keep (שמר) his commandments."
>
> Daniel 9:4 KJV

The human race expanded from Chapters 4 to 11 but continued to disobey God and rebel. God forgives them, but the disobedient pattern repeats. They drift further away from God, geographically and spiritually. Chapter 12 presents a turning point when Abram enters the story. He demonstrates obedience, which can be seen as the first step in repairing the relationship with God. Because of his faith, based on a promise, he obeys God's command to leave his home and faithfully migrate to the Promised Land.

> "Now the LORD had said unto Abram, Get thee out of thy country, and from thy kindred, and from thy father's house, unto a land that I will shew thee."
>
> Genesis 12:1 KJV

It is no ordinary adventure of several hundred miles into the unknown and danger. Despite the difficulties and perilousness, the migration becomes a spiritual journey of healing the relationship with God. The Book of Genesis closes with his grandson taking his family, the Israelites, to Egypt to escape famine.

The next book begins four hundred years later. The Israelites are strangers in the foreign land and suffer as enslaved people. God hears their cry and sends a prophet, Moses, to free them from the bondage of slavery and lead them back to the Promised Land. During this journey, God's teachings and commands are formally recorded in the remaining four books of the Pentateuch. The Pentateuch is known as the Torah in the Jewish tradition, which means the Law (Barton and Muddiman 2001, p. 12). The collections of laws are presented as having been revealed by God to Moses (and sometimes Aaron) to teach them how to live out their lives according to the creation purpose.

Creation is an expression of God's generous love. Man is made in God's image to spread such goodness. Therefore, love and compassion are the main themes of the Bible. In the Pentateuch, love and compassion are not only taught but are given as commands. One of the core commandments is "*thou shalt love thy neighbour as thyself*" (Leviticus 19:18 KJV). This commandment stands at the center of the central book in the Pentateuch and is the central commandment of the Pentateuch (Bamberger 1981, pp. 737, 889).

Special attention is paid to strangers among the vulnerable (the poor and needy, the widows and orphans). For example, in Leviticus 19:34, the command is explicitly extended to strangers: "*But the stranger that dwelleth with you shall be unto you as one born among you, and thou shalt love him as thyself.*" In addition, the proper treatment of strangers is comprehensively elaborated in the Pentateuch and further elaborated in the rest of the Bible. גור (sojourner) occurs 37 times in the Pentateuch, while גר and נכרי (both mean stranger) appears 68 times and 45 times, respectively. God often directly commands the Israelites to welcome and respect strangers while reminding them they were once foreigners in Egypt. For example, "*Love ye therefore the stranger: for ye were strangers in the land of Egypt,*" Deuteronomy 10:10 KJV. Strangers and foreigners are singled out from other vulnerable groups: the poor, the sick, the widows, and the orphans. When one welcomes a stranger, one is converted to a stranger who God welcomes back.

On the other hand, the Old Testament also includes warnings of curses and punishments for not welcoming strangers. For example, the prophet Jeremiah at the temple gate warns the people that if they continue to oppress the strangers, they will be driven to exile (forced migration): "*For if ye throughly amend your ways and your doings; if ye throughly execute judgment between a man and his neighbour; If ye oppress not the stranger, the fatherless, and the widow, and shed not innocent blood in this place, neither walk after other gods to your hurt: Then will I cause you to dwell in this place, in the land that I gave to your fathers, for ever and ever.*" (Jeremiah 7:5–7 KJV).

In the Old Testament, God's chosen people migrate from paradise to exile, from one place to scattering around the ancient world, from Ur to Haran to Egypt, then to the Promised Land, and from there to settle in Egypt for four hundred years, then, from Egypt back to the Promised Land and settle, then the exile and the return. Such history prepares them to be good hosts for foreigners. When they become good hosts, they will be welcomed by the ultimate host, and their primordial mission will be restored. The following section analyzes Jesus's explicit teaching on the mystical connection between these relationships.

## 4. The New Testament: The Restoration of Man—Welcoming Foreigners

Since the New Testament is written in Greek, we can only speculate how "shamar" is translated and discussed. To investigate how the understanding of "shamar" evolves, we switch the focus of this section to a group of people—the Samaritans.

*4.1. The Samaritans:* שומרונים/Σαμαρείτης — *A Brother or an Enemy*

> "*And he bought the hill Samaria* (שמרון) *of Shemer* (שמר) *for two talents of silver, and built on the hill, and called the name of the city which he built, after the name of Shemer, owner of the hill, Samaria.*"

<div align="right">1 Kings 16:24 KJV</div>

For the last few centuries, the term "Samaritan" has become synonymous with "someone who gives help to people who need it" (Samaritan 2022). However, the historical evidence for the exact origin of the Samaritans is scanty (Böhm 2020). It is beyond the scope of this work to present a detailed analysis. A brief history of the Samaritans is presented to provide the necessary perspective to understand their mindset and relationship with the Jewish people.

On the separation of Israel and Judah, Shechem's ancient city became the Northern Kingdom's religious center. King Omri transferred the political capital to his newly built city of Samaria. The city was the most luxurious capital of the kingdom until it fell in 722 B.C. (Kee et al. 1997, p. 529). "*That lie upon beds of ivory, and stretch themselves upon their couches, and eat the lambs out of the flock, and the calves out of the midst of the stall; That chant to the sound of the viol, and invent to themselves instruments of musick, like David; That drink wine in bowls, and anoint themselves with the chief ointments . . . .*" (Amos 6:4–6 KJV). It was burned to the ground by the Assyrian army. The Assyrian Empire deported the people of Israel to dispersed locations in the far north of the empire and sent colonists (five ethnic groups from Mesopotamia and the Levant) in to take their place. That is in line with the Assyrian military practice of transplanting the populations of captured territories to prevent future revolt. Then the Northern Kingdom of Israel disappears from history (Kee et al. 1997, p. 201).

However, Bergsma shows that, from the archaeological remains, the poorest of the Israelians—farm laborers—were left behind in exile and that some other populations fled to the safety of Judah (Bergsma and Pitre 2018). Hezekiah and possibly Josiah made forays into the north, inviting the Israelian population to worship at Jerusalem. The remaining population is then known as Samaritans, ethnically mixed descendants of the remnant Israelian population and the "*forced*" migrants.

Due to the lack of historical evidence, the exact terminology of Samaritans in Jesus's day is unclear. The first-century historian Josephus uses several terms for the Samaritans, which he appears to use interchangeably (Josephus 2011):

- peoples who live in the city of Samaria or the territory of Samaria (Cowley et al. 2021)
- peoples who are descendants of the forced immigrants
- peoples who declare themselves to be the true Israel and rightful heirs of the Land, claiming descent from Ephraim and Manasseh
- peoples who worship Yahweh accept only the first five books of the Old Testament as canonical and worship on Mount Gerizim instead of on Mount Zion

However, most historians and theologians agree that there is intense enmity between them and the inhabitants of Judah (Kee et al. 1997, p. 201). When Jesus was a child, the Samaritans defiled the Holy Temple in Jerusalem by strewing dead men's bones into the sanctuary during the Passover around A.D. 6 and 9 (Jeremias 1963, p. 204). For pious Jews, there is no greater offense than defiling the Temple. Therefore, there are no greater enemies than the Samaritans.

In the days of Christ, the hatred was so fierce that the Jews bypassed Samaria as they traveled between Galilee and Judea. They went an extra distance through the barren Land of Perea on the eastern side of the Jordan to avoid going through Samaria (Kee et al. 1997, p. 529). Calling someone a Samaritan was considered an insult (John 8:48). However, Jesus rebuked his disciples for their hostility to the Samaritans (Luke 9:55–56) and preached to the Samaritans (John 4:40–42).

### 4.2. Luke 17:11–19—The Grateful Samaritan Leper, the Stranger (αλλογενής)

*"There are not found that returned to give glory to God, save this **stranger**."*

Luke 17:19 KJV

On his final journey to Jerusalem, Jesus passes through the midst of Samaria, ten lepers, seeing him, raise their voices and say, "*Jesus, Master, have mercy on us*" (Luke 17:13 KJV). Jesus heals all ten, telling them, "*Go shew yourselves unto the priests*" (Luke 17:14 KJV). All left, but only one, a Samaritan, eventually returned to glorify God "*with a loud voice*" and thank him. Then Jesus tells him, "*Arise, go thy way: thy faith hath made thee whole*" (Luke 17:19 KJV).

This miracle is unlike most others since the healing itself is not emphasized as much as the reaction to it. Curing from leprosy is a blessing no different from wealth, might, and knowledge. The Samaritan leper does not consider everything as their due but as a gift. His relationship with society is restored through the priests. However, Jesus says his faith makes him whole (instead of Jesus's power making the leper whole). Faith entails opening the leper's heart to God's grace; it means recognizing that everything is a gift and everything is grace. Because of his reaction to God's blessing, he is saved. The Greek word σέσωκέν (sésokén) is translated as "hath made thee whole." The root is σῴζω (sózo)," which means "save, heal, preserve, and rescue.".

It is also necessary to investigate the fact that he is a Samaritan. First, because leprosy is such a severe illness, the Jewish and Samaritan lepers overcome the hatred between their ethnic groups and form a community of outcasts. If they are all lepers, what does it matter? When they meet Jesus, they are in excruciating physical and emotional pain. After Jesus heals them, the Samaritan believes in his healing power and has the courage and wisdom to go to the priests, who most likely would not welcome a Samaritan. So, the story is more than that Jesus is expecting and excited to receive gratitude. He is overjoyed to see the faith of the Samaritan. Coincidentally, in the KJV, it is translated as "save this stranger (αλλογενής)." The Greek word αλλογενής appears only once in the Bible. It means the stranger who dwelt within the land. It is the same Greek word used in the inscription around the temple courts, allowing access only to Jews but to αλλογενής (Bühler 2023).

After the story of the ten lepers, Luke ends the chapter with "The Coming of the Kingdom." When the Pharisees demand Jesus to reveal when the Kingdom of God would come, he replies, "*The kingdom of God cometh not with observation: Neither shall they say, Lo here! or, lo there! for, behold, the kingdom of God is within you*" (Luke 17:21 KJV). The Samaritan leper could be seen as an example of those who are in the Kingdom of God. The other nine lepers receive God's blessing and are healed physically but do not reach the Kingdom. They could be very grateful to Jesus but might have to rush home to see their families. Sadly, the healing does not teach them anything about who Jesus is and where the Kingdom of God is. So, they are not made "whole"/"σῴζω." Indeed, the Samaritan leper has more reasons to go home first as his family is probably in Samaria. However, he knows that there is someone or someplace more at hand. "*The time is fulfilled, and the kingdom of God is at hand*" (Mark 1:15 KJV). Therefore, he "*hath chosen that good part, which shall not be taken away from*" him (Luke 10:42 KJV)[1].

The story resembles another Old Testament story about a foreigner leper—Naaman the Assyrian, in 2 Kings 5:1–27. Naaman is more than a foreigner. He is not just an enemy but a commander of the enemy's army that will wipe out the Kingdom of Israel. Three aspects parallel with the Samaritan leper are worth mentioning. Elisha is in Samaria when Naaman comes for help. "*Would God my lord were with the prophet that is in Samaria*" (2 Kings 5:3 KJV). Naamen pays two talents, "*And Naaman said, Be content, take two talents.*" Naamen commits to serving God only and brings him home. "*Shall there* not *then, I pray thee, be given to thy servant two mules' burden of earth? for thy servant will henceforth offer neither burnt offering nor sacrifice unto other gods, but unto the LORD*" (2 Kings 5:17 KJV).

By opening their hearts to God, the Samaritan Leper and Naaman convert inwardly and are healed from the brokenness of the human condition. They become keepers of the

Law and serve God. Even though they are foreigners, they are not afraid to march toward their newfound mission and prepare themselves to collect their eternal rewards. That is why Jesus can confidently tell him, "*Arise, go thy way,*" as his way becomes God's way.

*4.3. John 4:4–30—The Samaritan Woman and the Stranger*

> "*And many of the Samaritans of that city believed on him for the saying of the woman, which testified, He told me all that ever I did.*"

<div align="right">John 4:39 KJV</div>

The story of the Samaritan woman is one of the most iconic encounters in the New Testament (Barton and Muddiman 2001, pp. 967–69). She is never named, yet her encounter with Jesus is the longest between him and any other individual in the Gospel of John—almost 40 verses. Despite representing the lowest of the low—a female in a society where women are both demeaned and disregarded, a race traditionally despised by Jews, and a social outcast living in shame even in her community—she receives and brings others eternal life.

Jesus demonstrates a critical element in *keeping* others. Jesus met the woman in Samaria, a place that Jews avoid. Again, he is the αλλογενής there. "*Being wearied with journey . . . *" (John 4:6 KJV), he was tired after walking for six hours (Groody 2022, p. 188). "*It was about the sixth hour*" (John 4:6 KJV); it is midday, the hottest time of the day. Women collect water at other times to avoid the heat. But the Samaritan woman has to come at this time because she is an outcast. Water drawing is a communal rather than individual activity (Dunn and Rogerson 2003, p. 1171). It shows that the community does not accept her. However, despite all this, Jesus reaches out to her to transform her.

When she hears "*I that speak unto thee am he*" and realizes that she is blessed with an encounter with the "*Messias,*" she "*left her waterpot, and went her way into the city, and saith to the men.*" She does not keep the blessing all to herself. She does not ask if she would be welcome in the city or if people would listen to an unlearned woman, especially an outcast. She is transformed from a despised reject to a luminous evangelist sharing her encounter with the eternal life: "*He told me all that I ever did.*" She hurries without delay to proclaim the good news to the nearby village and does not hold back in sharing her shady past if it could help others restore their relationship with God. She becomes another keeper of the Law and serves God without reservation. The reward for her work is that Jesus stays in the city with the Samaritans for two days.

The complexity of the immigration issue could become a barrier for those who have the desire to help. One could feel that more skills and commitments are required to support a migrant than a neighbor, such as languages, time commitments, safety, etc. "*Fear not*" or similar appears in the KJV 113 times. In some verses, it is paired with "*Fear thou not; for I am with thee*" (Isaiah 41:10). There is no required qualification or skill when helping a brother or sister in need. *Keeping* others is a universal call not just to the learned, rich, and mighty.

*4.4. Luke 10:30–37—The Good Samaritan, the Ανθρωπος, and the Transformation*

> "*Then said Jesus unto him, Go, and do thou likewise.*"

<div align="right">Luke 10:37 KJV</div>

The Parable of the Good Samaritan is one of the most famous stories in the Bible (Sloyan 1983). Before the parable, a particular lawyer tempts Jesus by asking him, "*Master, what shall I do to inherit eternal life?*" (Luke 10:15 KJV). After quoting the Old Testament, "*Thou shalt love the Lord thy God with all thy heart, and with all thy soul, and with all thy strength, and with all thy mind; and thy neighbour as thyself,*" (Luke 10:27 KJV), he asks "*And who is my neighbour?*" (Luke 10:29 KJV). This question shows that he does not want to view some people as his neighbor, meaning he does not want to treat others as himself (Dunn and Rogerson 2003, p. 1126). Given that he is a scholar of the Law, it would be sensible to assume that he is Jewish and likely does not harbor respect for Samaritans (Barton and Muddiman 2001, p. 942). Accordingly, Jesus seeks to shift his perspective by making the

Samaritan the story's hero, ultimately calling him to recognize that Samaritans must be treated as himself.

This story is universally called the Parable of the Good Samaritan. However, Jesus does not explicitly call it a parable. Ratzinger comments that the story is perfectly realistic because such assaults are a regular occurrence on the road to Jericho (Ratzinger 2007, p. 196). Since the Samaritan does not live in Jericho or Jerusalem, he is likely to be a merchant traveling by since he leaves after putting the man in the care of the innkeeper, whom he apparently knows. He has more reason not to stop since he and his beast could carry valuables.

On the other hand, for those who read the Septuagint, their minds switch back to the Book of Genesis when they hear the word ἄνθρωπος (ánthrōpos). It is the word used for the "*man*" in the Creation narrative. He represents humanity, and the story becomes man's mission.

Seeing a man in desperate need could be a blessing. It can be seen as a "low-hanging fruit" opportunity to share God's gift. The priest, the Levite, and the Samaritan are presented with the same blessing and opportunity. The same Greek word ἰδών (idṓn, "having seen") is used for each of them. Accordingly, the story can be rephrased thus: "God sends three men to be a neighbor of his ἄνθρωπος. Only a man comes back with this ἄνθρωπος.".

However, since the ἄνθρωπος is half dead, there is no reason to stop because they neither have the medical skills nor the medical supplies to assist him. They might make the situation worse. As with the Samaritan woman, heeding the universal call of helping a stranger regardless of the condition, the Good Samaritan does not ask the questions of whether he is welcome, qualified, and capable of saving the ἄνθρωπος. He just acts even though his effort to "shamar" could be fruitless, even rejected, delaying his business dealing and putting him in danger.

> *"Which now of these three, thinkest thou, was neighbour unto him?"*
>
> Luke 10:36 KJV

More importantly, Ratzinger recognizes that the discordance between the lawyer's question and Jesus's answering question demands a deeper investigation (Ratzinger 2007, p. 197). A more direct reply should be, "Is the man or the Samaritan, your neighbor?" Then, the parable simply becomes a story of helping another—kindness. Jesus answers by reversing the question and showing the excellent work of the Samaritan and, most importantly, his inward conversion. The issue is no longer who is the lawyer's neighbor. But instead, who can be a neighbor, and how. The Greek word ἐσπλαγχνίσθη is translated to "*had compassion on*", which diminishes its original vitality. The root of the word is σπλάγχνα (splánkhna), which means the internal organs, especially the bowels. It means "to be moved" in one's bowels—for the bowels were thought to be the seat of love and pity in Greek—or, as we would say now, to feel from the gut.

Ratzinger suggests that the Hebrew/Aramaic word Jesus uses could be רחמים (rahma) (Strong 1890, H7356), which is translated in the KJV as mercy, compassion, womb, bowels, pity, damsel, and tender love. Similarly, it uses the internal organs to signify the emotion of the human being. However, it goes beyond with a virtue par excellence—motherly love. Ratzinger calls it a mother's love for her womb. When a woman conceives a child, she is transformed into an image of God and assigned a title—the mother of a new creation. Similarly, when the Samaritan "ἰδὼν ἐσπλαγχνίσθη", he is transformed into a neighbor and bestowed with a title—the keeper of the Ἄνθρωπος. Another possible Hebrew word could be חסד (hesed) (Strong 1890, H2617), which also means mercy and kindness. חסד appears 250 times in the Old Testament and is the fourth most used noun after God, LORD, soul, and Land. God names himself with חסד (Exodus 34:6–7) "*and shewing mercy unto thousands of them that love him and keep thy commandments*" (Exodus 20:6) because חסד is what is needed to see an enemy in desperate need as a neighbor.

Seeing the ἄνθρωπος in such a state is a blow that strikes him viscerally[2], touching his soul (Ratzinger 2007, p. 197). Struck in his soul by the lightning flash of mercy, he himself

now becomes a neighbor, transcending any political or cultural barriers and even danger. His deep compassion converts his heart: the issue is no longer whether the other person is a neighbor to him; he has to become a neighbor. He becomes like someone in love whose heart is open to being shaken up by another's needs. Then he finds his neighbor, or better yet, then he is found by the other person. The Samaritan goes beyond simply doing a good deed out of compassion; he sees the other person "as himself." In the Greek text, "ἰδών" and "σπλαγχνίζομαι" go together as if they are a single act. "ἰδών σπλαγχνίζομαι" is both a blessing and a sign of the restoration of his relationship with God.

σπλαγχνίζομαι is not a common Greek word and appears 12 times in the New Testament, either describing Jesus, calling him, or used by him. Jesus "*saw the crowds*" (Matt 9:36), "*saw a multitude*" (Matt 14:14), "*have compassion on the multitude*" (Matt 15:32), "*had compassion on the two blind men*" (Matt 20:34), "*moved with compassion and touched the leper*" (Mark 1:41), "*saw much people*" (Mark 6:34), "*have compassion on the multitude*" (Mark 8:2), "*saw her (the widow of a dead man)*" (Luke 7:13). In addition, Jesus uses the word in his parables to describe the lord of the unforgiving servant, the father of the prodigal son, and the Good Samaritan. The boy with an evil spirit said, " . . . *have compassion on us*" (Mark 9:22). Σπλαγχνίζομαι is a divine emotion. One can only receive it if one sees others with an open heart and is ready to be converted. The Samaritan becomes the hero of Jesus's story because he allows God to convert him from the Samaritan, an enemy of the Jews, to the Good Samaritan, a keeper of brothers and sisters of any ethnic group, even an archenemy.

Migrants usually come with different languages, cultures, and religions. Engaging a migrant appears more challenging than doing local charity work. In addition, there is always a possibility of getting rejected. If the Samaritan chose to pass by on the other side, he would have missed the opportunity to be a keeper of the Law and serve God.

In conclusion, the priest, the Levite, and the Samaritan are given the same blessings to see a neighbor and the opportunity to become one. The priest and the Levite choose to go out of their way to avoid helping ("ἀντιπαρῆλθεν", "passed by on the other side" Luke 10:31,32 KJV) and let the blessing pass by. Only the Samaritan sees a neighbor (as himself) and allows his heart to be changed. He takes on the mission of becoming Jesus and trying to restore the brokenness relationship of the Άνθρωπος. Because of that, he "*inherits eternal life.*".

### 4.5. The Three Samaritans and the Unconditional Welcome

The three Samaritans are never named but become famous, especially the Good Samaritan. Hospitals[3], schools, churches, and charitable organizations[4] worldwide are named after him[5]. "Samaritan" becomes an adjective that praises a generous person and sets a new standard for how one should treat another. How did the Samaritans in the Bible become such an admired group of people? In the Oxford Dictionary, Samaritan means "a person who gives help and sympathy to people who need it", which summarizes the characters discussed in the last three sections and their encounters with God—the transformation of the meaning from "fool" and "enemy" to "generous" and "brother." The general theme is they receive God's blessings with an opportunity of an encounter. Some choose to use the opportunity to restore their relationship with God through an inward conversion, pursue the newfound mission, and be ready for the final reward. In contrast, others prefer to "*pass by on the other side*" or not to be seen, as summarized in Table 1.

On the other hand, as discussed in Section 3.2, man is created to be the mediator between God and creation and the vice-regent of all creation. The keeper role is lost because of the First Sin. However, God continues to teach and restore man to the primordial role through the Law and the Bible. Figure 2 shows that the Christian teaching of unconditional welcome to anyone is part of man's original design and an essential theme of the Bible. Through the theme of "shamar" in the Bible, man can choose to journey back to the Garden with God through an inward conversion by seeing and welcoming a foreigner.

**Table 1.** The Inward Conversion.

| Foreigners | Blessings | Restoration with God? | Mission | Reward |
|---|---|---|---|---|
| The nine lepers | *"they were cleansed"* | *"Go shew yourselves unto the priests"* | — | — |
| The Samaritan leper | *"they were cleansed"* | *"Go shew yourselves unto the priests"* *"with a loud voice glorified God"* | *"Arise, go thy way"* | *"thy faith hath made thee whole"* |
| Naaman the Syrian | *"his flesh came again like unto the flesh of a little child"* | Naaman starts to address himself as *"thy servant"* | *" . . . will henceforth offer neither burnt offering nor sacrifice unto other gods, but unto the LORD"* | *"he was clean"* |
| The Samaritan woman | *"I that speak unto thee am he"* | *"many of the Samaritans . . . believed on him for the saying of the woman, which testified"* | *" . . . left her waterpot, and went her way into the city, and saith to the men"* | *"a well of water springing up into everlasting life"* |
| The Ἀνθρωπος | *"bound up his wounds . . . and took care of him"* | *"brought him to an inn . . . "* | *"pouring in oil and wine"* | *"Take care of him; and whatsoever thou spendest more,"* |
| The Priest | *"ἰδὼν"* | *"ἀντιπαρῆλθεν"* *"passed by on the other side"* | — | — |
| The Levite | *"ἰδὼν"* | *"ἀντιπαρῆλθεν"* *"passed by on the other side"* | — | — |
| The Good Samaritan | *"ἰδὼν"* | *"σπλαγχνίζομαι"* *"was moved with compassion"* | *"neighbour"* | *"inherit eternal life"* |

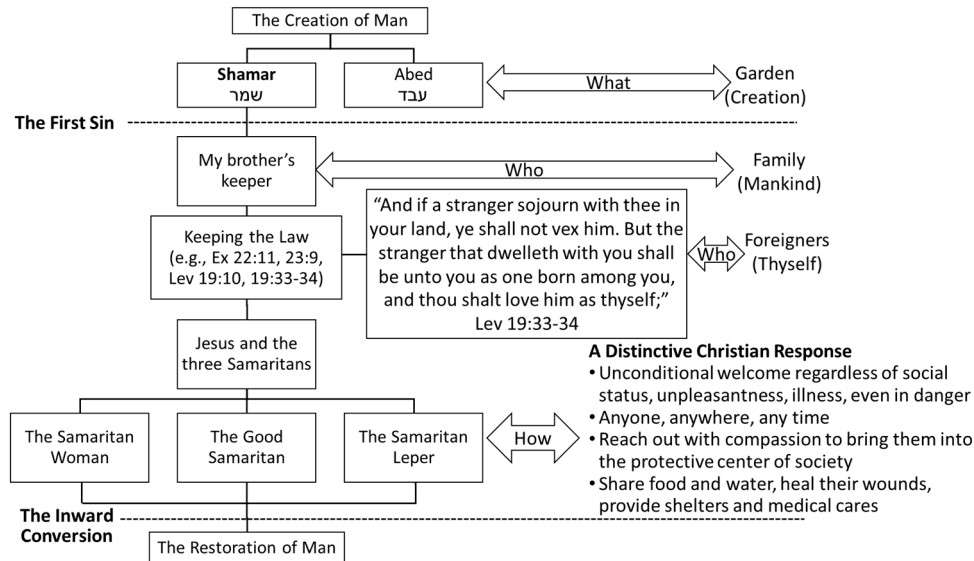

**Figure 2.** The Theme of "Shamar.".

## 5. Discussions: Application of a Mindset Guided by "Shamar"

> *"What the Bible says is uncontroversial, but how it is to be applied in practice in daily life raises questions that really are controversial among scholars and in everyday life."*
>
> Joseph Ratzinger (Ratzinger 2007, p. 195)

With an understanding that the theme of "shamar" can be a guide for how action must be taken concerning the issue of migration, it is now possible to more concretely consider how this mindset plays out in the real world. This framework calls for personal action: people must open their hearts and be ready to be radically transformed: effective action

must begin with oneself. None of the three Samaritans are called to enact far-reaching political or societal changes. They are praised for their efforts, from sharing a cup of water to caring for a wounded victim. That is not to say that wide-reaching change will not occur. On the contrary, like the three Samaritans, our transformation becomes an example and calls for others in the community to become inspired themselves. Here are some key points:

- **The eye of a migrant**: The Good Samaritan story shows that it is more important to see with the eyes and heart of a neighbor than to find a neighbor to help. Only when one becomes a neighbor does one see a neighbor or is founded by one. Eventually, one sees the neighbor as oneself. Similarly, when one sees and thinks in the mindset of a migrant, one would know who to help and how to help. After all, humanity is an immigrant on the earthly pilgrimage.

- **The primordial mission**: Helping a brother or sister in need is a part of God's commands in the Garden of Eden. The mission has not changed from Adam to Jesus. The lawyer's question about inheriting eternal life can be rephrased as "What is the purpose of man?" The mission reemerges when the heart is radically transformed. The Good Samaritan does not ask if what he does would earn him eternal life. He goes out of his way to help because he sees a Creation of God in desperate need.

- **Good Samaritan Laws (Duty to Act)**: Many countries write laws to impose on their citizens to act like the Good Samaritan and give aid of any sort to a victim. For the context of this research, these laws do not coincide with the teaching of the Bible. The work of mercy does not earn eternal life. It is the transformed heart. That is why Jesus does not answer the lawyer's question directly. The focus is not on who I am required to love, but on who I am.

- **The Good Samaritan vs. the demonstration effect**: In most artworks of the parable, the priest, the Levi, and the Samaritan travel behind one another[6] (Green 1997). When the Levite walks by, he sees that the priest does not stop. If even a priest does not help, he probably should not. As described in sociology, the behavior could result from the demonstration effect (Ahmed 2017). It refers to an individual's behavior caused by observing others' actions and consequences. When the Samaritan walks by, he probably overcomes the demonstration effect and focuses only on the wounded man, not on who does not stop. Many of the migration issues are controversial and confusing. Many choose to avoid them. It is tempting to follow the crowd and shun the problems. However, a better way is to focus on the wounded and hurt than those who pass by. One should focus on Jesus and the Good Samaritan rather than the powerful and mighty. If Adam focuses on God and keeps the Garden, he probably will not have time to entertain the serpent. If Cain concentrates only on his own offering, he probably will not have time to become jealous.

- **The Good Samaritan vs. the Bystander**: On the other hand, one is less likely to offer help if one is in a large group—the bystander apathy. The apathy could also apply to companies and nations. In social psychology, the phenomenon is called the Diffusion of Responsibility. When people are in a large group, the responsibility to take action is diffused through the entire group (Darley and Latane 1968, Bystander Intervention in Emergencies: Diffusion of Responsibility), (The Good Samaritan Effect (Definition + Examples) (2023)). Social psychology researchers from Princeton University conducted an experiment to study the impact of the story of the Good Samaritan on helping behavior (Darley and Batson 1973). The subjects were asked to complete one task, walk from one building to another and work on a second task. While in transit, the subjects passed a slump "victim" planted in an alleyway. Their behavior was observed and measured. The results showed that if other people had been on campus as the subjects noticed the victim, they might have been less likely to stop. For some subjects, the first task was reading and writing about the Good Samaritan, and the second was presenting it. The results showed no correlation between religious types and helping behavior. Some "more religious" subjects even stepped over the victim.

That is, knowing the Good Samaritan did not have any influence on helping behavior. The only variable that showed some effect was "religion as a quest." Only those who considered religion a quest were more likely to care for a brother in desperate need. Because of the scale of the migration crisis, it is easier to assume that only charitable organizations and nations with the necessary human and financial resources are more capable of handling it. Maybe the United Nations and the World Bank. What could one person do? However, the Samaritan woman does not feel that a lowly social outcast could do nothing to help her fellow villagers and serve God.

Many obstacles, such as the demonstration effect, bystander apathy, or other political and social barriers, discourage potential Samaritans from engaging people in desperate need, especially migrants. That is why conversion from the inside is essential.

## 6. Conclusions

*"This is my commandment, That ye love one another, as I have loved you."*

John 15:17 KJV

This paper studies the theme of "shamar" in the Bible and how it can be used to understand and analyze the issue of migration. The research is divided into two parts: (1) investigating the meaning of "shamar" in the Old Testament and the relationship with the mission of man; (2) analyzing the stories of the three Samaritans and their transformation. Figure 3 illustrates the approach taken to understand the Christian's role in the phenomenon of migration. The study begins with an exegesis of the Creative Narrative to understand the purpose of man. Man is created to be the vice-regent of all creation, not just the neighbors, and to be the keeper of neighbors and strangers. Unfortunately, man loses his priestly role in the Garden of Eden and has to be radically transformed to restore the original mission. The spiritual journey of man back to God can be traveled through an inward conversion. Only then, one knows and acts with the correct disposition to an immigration.

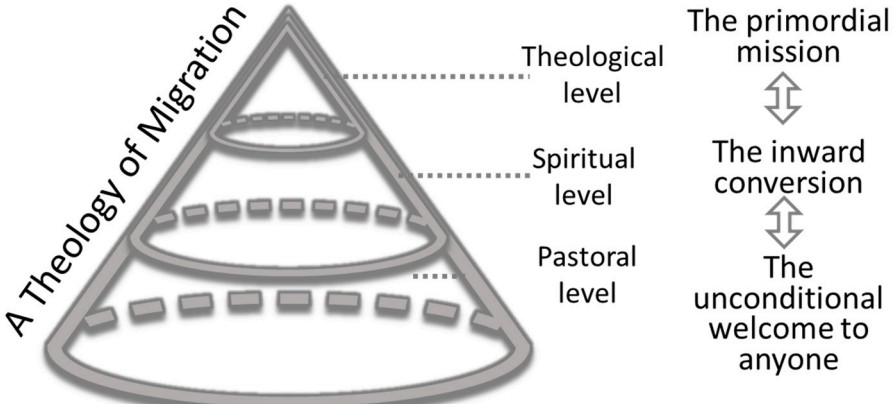

**Figure 3.** A Theology of Migration through the Biblical Theme of Shamar.

The examples set by the three Samaritans form a framework for living out the mission and command of "shamar." Following this framework, one is transformed into an outpouring of neighborly love. The Good Samaritan is a symbol of Jesus, a neighbor to the migrants. He sees them as neighbors, cares for them even if his life could be endangered, takes them to an inn before handling his own business, and finds them a trustworthy keeper before leaving. That is what "shamar" means. As the migration crisis continues to grow, truly and effectively addressing it requires a radical transformation, as outlined through the experiences of the three Samaritans. Each becomes transformed through an intimate encounter with Christ, either personally or through the poor and vulnerable. This compels each to act and care for their neighbors. As one seeks to imitate their examples, one must be open to this conversion and the actions that follow.

**Funding:** This research received no external funding.

**Data Availability Statement:** Not applicable.

**Acknowledgments:** The author is very thankful to Kate Motroni-Fish, William Bonnell, John Donald, and Daniel Groody for their guidance and support.

**Conflicts of Interest:** The author declares no conflict of interest.

## Notes

[1]  Note that the story of "Martha and Mary" is immediately after the Parable of the Good Samaritan.

[2]  The Latin root of "visceral" is "viscera", which also means "the internal organs of the body."

[3]  Over forty general hospitals in the American Hospital Assoiacions are named after Good Samaritan.

[4]  International humanitarian aid organizations from Samaritans's feet -to Samaritan's Purse (Knowing God 2023).

[5]  According to Real Yellow Pages, more than 3000 businesses in the U.S. have the word "Samaritan" in their names (Samaritan 2023).

[6]  For example, Domenico Fetti's The Good Samaritan, Jan Wijnants' Parable of the Good Samaritan, Giacomo Conti's Il buon smaritano, and Van Gogh's the Good Samartian.

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
