# Peer review of "An Understanding of Christians’ Roles in Human Migration through the Biblical Theme of Shamar: From Genesis to the Good Samaritan"

_religions, doi:10.3390/rel14050600_

Round 1

Reviewer 1 Report

I have no objections.

Author Response

Dear Interviewer,

Thank you very much for your comments. My paper has been revised and resubmitted for your consideration. Here is my response to your comments:

  1. Clarifying my method: I have added a new section (Section 2) to describe the research method. Section 2 describes the approach – Narrative-Critical Biblical Reflection. Section 2.0 explains the framework used for this work.
  2. Making my objective more specific: I have added a new section (Section 1.0) to review the existing literature on the topic and identify an opportunity for an original contribution. At the same time, the research objective and question are presented.
  3. Organize my paper: Section 1 is rewritten completely. The old section 1.1 and old table 2 are removed. Most sections are shortened to make the text more readable.

Again, thank you very much for your time and consideration.

Sincerely.

Reviewer 2 Report

Title: A Theological Framework for Understanding Migration through the Biblical Theme of Shamar: From Genesis to the Good Samaritan

While this study has taken an innovative approach to make connection between migration and Religion, the article lacks substantive contents.

First, the study needs to discuss how they are making this connection between Shamar and migration. Basically, the author have to clearly state the method they are using. From lines 100 to 113 they have attempted to partly mention the method that they used to do this method, yet it is absolutely unclear and lacks sufficient information.

The author also mentions the objective of the paper very briefly, although it is stated at a right spot. The reader of this manuscript did not have to wait too long to understand the objective. However. The objective itself is very scanty and insufficient.

The author needs to organize the paper better. The way it is now, reads completely disorganized. For example, throughout the paper the author mentions how some discussions are beyond the scope of this paper. The author must mention this at the onset whatever the discussion is that author thinks is beyond the scope of this paper.

The table contents are meaningless, What the author needs is the table that illustrates themes highlighting the thematization of the migration with Samaritan. The paper suffers from empirical rigor that makes this paper not eligible for publication in the way it is now. Also the study needs a lot more content from existing literature.

The whole paper needs rewriting. I am afraid that I will not accept the paper. 

Author Response

(The authors gave the same response as above.)

Reviewer 3 Report

I really liked this splendid paper. The analysis of the place of the Samaritans in Christian thought, and especially in Razinger's theology, its' ramifications on the treatment of immigrants, and finally the interesting analysis of the Hebrew verb "Shamar", all merges into a very convincing argument.

Author Response

(The authors gave the same response as above.)

Round 2

Reviewer 2 Report

Accept as it is.